# Security Assessment of Taiwan Solid Wood Product Supply

**Chyi-Rong Chiou [1]** , **Wei-Hsun Chan [1,2]** , **Meng-Shan Wu [2]** and **Jiunn-Cheng Lin [3],***

1 School of Forestry and Resource Conservation, National Taiwan University, Taipei 106, Taiwan;
esclove@ntu.edu.tw (C.-R.C.); chchan0505@gmail.com (W.-H.C.)
2 Division of Forestry Economics, Taiwan Forestry Research Institute, Taipei 100, Taiwan;
wumengshan@tfri.gov.tw
3 Taiwan Forestry Research Institute, Taipei 100, Taiwan
\* Correspondence: ljc@tfri.gov.tw; Tel.: +886-2-2303-9978

**Abstract:** Taiwan is highly dependent on imports of solid wood materials. In the past ten years (2009–2018), it imported raw materials for solid wood products from 117 countries. Therefore, the diversity of raw material sources is a serious concern. The purpose of this research is to evaluate the risks of solid wood product supplies. The dispersion and concentration of supply sources are the measures. The SWI and HHI models are used to calculate the six major imported solid wood products. The results show that from the beginning of 2009 to the end of 2018, wood chips and pellets with the highest average annual import quantity had the lowest average performance in SWI and with the highest average HHI value, which indicates that wood chips and pellets are the riskiest items among the 6 solid wood products. While the sawn wood has the highest average SWI value and the lowest HHI value, it offers the smallest supply risk.

**Keywords:** material security; SWI indicator; HHI indicator

## 1. Introduction

Wood is a necessary raw material for livelihood and the development of the wood industry, and it is also one of the most important resources for economic development. In the 1970s, the domestic wood production of Taiwan exceeded 1 million m$^3$. With the awareness of environmental protection increasing and the trend toward homeland security, Taiwan's forestry policy goals have been transformed since 1991 to be aligned predominantly toward conserving forest resources. Taiwan's forest management plan specified a comprehensive ban on clearing natural forests, and the annual harvest is limited to less than 0.2 million m$^3$ [1]. The annual domestic wood production has declined, yet the domestic demand for forest products has not decreased. Manufacturers can only rely on imports to stabilize the availability of a number of raw materials for continuous production.

In recent years, the annual domestic forest harvesting quantity is only about 40,000 to 60,000 m$^3$, while the total annual domestic demand for wood materials is about 4 to 6 million m$^3$, meaning that more than 99% of solid wood products are obtained through imports, and the wood self-sufficiency rate is less than 1% [2,3]. The forest industry-related manufacturers of Taiwan can only import raw materials for sustainable and stable production. The first priorities for timber manufacturers when selecting raw materials are "stable supply source," "reasonable price," "sufficient supply quantity" and other factors; therefore, with low domestic wood supply, manufacturers naturally rely on imported wood [4]. In the scenario of low flexibility in raw material prices for high demand, manufacturers have to fully understand the sources of imported timber in order to improve their competitiveness in the face of changes in business conditions and exchange rate fluctuations [5].

From the perspective of the sustainable management of national resources, forests are renewable resources that should withstand change and continue to be productive over time. Foreign studies have pointed out that the level of wood self-sufficiency is closely related to the sustainability of the forest [6]. The extensive use of imported wood

and forest products in Taiwan has caused the weakening of domestic wood production, leading to the self-sufficiency rate being less than 1% for several years, which is an unusual employment of resources. Furthermore, in recent years, due to the increasing awareness of worldwide forest environmental issues, the international community has been focusing on reducing global warming, conserving biodiversity, combating illegal logging, promoting forest certification, and reducing greenhouse gas emissions from deforestation and forest degradation (REDD +, reduction in emission from deforestation and forest degradation), which has a major impact on the global forest product trade market accordingly imposing stricter rules on the import and export of timber and forest products. This will inevitably cause the price of timber to rise while the source and quantity decrease, thereby affecting the market structure, supply and demand, and impacting the development of Taiwan's wood-related industries.

In light of this, one of the conclusions of the National Agricultural Congress, held by the Council of Agriculture (COA) in 2018, is to ensure domestic sources of wood materials, the proper use of domestic forest resources, and effective promotion of the national forestry industry. In response to structural changes in the international forest products market in the future, Taiwan will reduce its dependence on imported timber and propose a policy goal of achieving a self-sufficiency rate of 5% of domestic timber production within 10 years.

Most of Taiwan's timber comes from imports, just like Taiwan's energy demand, which also depends on imports. The dependence on energy imports exceeds 97% [7]. Since both raw materials and energy rely highly on imports, the stability and risk of the raw material supply are paramount; hence, the consideration of supply security and national security is also involved. The diversity of wood-importing countries is an important element to ensure energy security [8]; it helps respond to internal and external environmental changes and impacts, and can reduce the vulnerability of supply disruptions from a single source. In terms of energy, the higher the proportion of imports or the reliance on a single import, the greater the risk of supply security; to keep energy, the environment and sustainable development in equilibrium, the diversity and localization of the energy system is the best approach while contributing to energy security [9].

Diversity indicators have been widely used by many scholars in various fields of research, among which the SWI (Shannon–Wiener indicator) and HHI (Herfindahl–Hirschman indicator) models are most commonly used and discussed [8]. In the case of energy security, in particular, much international related research uses these two indexes for measuring security. As stated by Chalvatzis and Ioannidis [10], their research on the diversity and dependence of EU energies used the SWI indicator to measure diversity of energy, and the HHI indicator for energy dependence. They also explored the connection between index values and the utilization rate of renewable energy. Correspondingly, these two indicators are also commonly used in the diversity of renewable energy in Asia [11,12]. In addition, Le Cog and Paltseva [13], as well as Rubio-Varas and Beatriz [14], respectively, conducted their research calculating the composite index of energy security and exploring the problem of energy concentration by applying an extended formula. Given that, there has been sufficient literature on the application of energy security indicators. Wang et al. [15] used the SWI to assess the risks in China's timber imports. There are, however, few studies related to timber; the literature on the security of timber supply is insufficient. Drawing upon the core concept addressed by the aforementioned research on related energy security, this study provides a conceptual definition as the point of departure that the key to promoting the security of timber supply lies in the diverse sources of solid wood products, using the research by Chalvatzis and Ioannidis [10], Lo [11], and Tufail et al. [12] as the main references. Based on the concept of wood supply security, and the conditions of Taiwan's domestic solid wood product imports, this study uses SWI and HHI as assessment models to analyze the import sources and the quantity of domestic solid wood products. However, SWI and HHI do not reflect the disparity dimension of diversification. They only address the number and proportion of wood product resources [16,17], but measuring the

diverse sources of solid wood products in order to assess the security of wood can be used as a reference for the development of national wood supply and demand strategies.

## 2. Materials and Methods

### 2.1. Taiwan's Major Solid Wood Product Imports

Adopting Food and Agriculture Organization (FAO) classification standards and definitions [18], the centerpiece of the research method used in this study is the Joint Forest Sector Questionnaire (JFSQ) developed by the International Tropical Timber Organization (ITTO), United Nations Economic Commission for Europe (UNECE), and the FAO to collect trade data of wood and forest products from each country. Based on the import and export records of the customs administration under the Ministry of Finance, the statistical data of wood and forest products analyzed are presented in units of solid wood equivalent volume. Solid wood products can be divided into 6 major categories using the FAO classification, including roundwood, wood charcoal, wood chips and pellets, wood residues (other wood chips, bark, sawdust), sawn wood, and wood-based panels. Roundwood includes wood fuel (including fuelwood) and logs for industrial use, and wood-based panels including veneer, plywood, pellet board and fiberboard.

### 2.2. Assessment Model

Regarding the importing countries and quantities of different types of wood in Taiwan, two evaluation methods, the SWI (Shannon–Wiener indicator) and HHI (Herfindahl–Hirschman indicator), were used in order to assess the security of solid wood product supply. The criteria of comparison of this study are based on the aforementioned principle: the higher the degree of data dispersion, the smaller the risk; conversely, the higher the concentration, the greater the risk.

#### 2.2.1. Shannon–Wiener Indicator (SWI)

SWI is an ecological indicator of biodiversity, and it also includes the concept of entropy. Related studies apply it to the supply of raw materials, that is, to calculate the randomness or confusion of data. If the number of importing countries is large and the average import quantity of the item is large, the proportion of the total is similar, resulting in a large SWI value indicating that the data are scattered and random. On the contrary, if the gap between the items is large, the SWI value is smaller and indicates that the data are less random [15,19].

This indicator first calculates the ratio of the annual import of solid wood products to the total import of solid wood products. Using this ratio $P_i$ to take the natural logarithm $lnP_i$, then multiplying the two and converting them into positive values $P_i \times lnP_i$, and lastly, totaling the $P_i \times lnP_i$ values of each country to calculate the sum of the six major annual projects $P_i \times lnP_i$, the calculation result indicates that each year has 7 SWI values, including the sum of 6 solid wood products and all raw materials. The formula is as below:

$$\text{SWI} = -\sum_{i=1}^{n} (P_i \times lnP_i) \tag{1}$$

where $P_i$ = the proportion of solid wood products in total imports.

#### 2.2.2. Herfindahl–Hirschman Indicator (HHI)

The HHI model is a statistical method developed by Hirschman [20] and Herfindahl [21] to calculate market concentration for analyzing market power, and to calculate the percentage of the manufacturer's market share: the higher the percentage of sales against total sales, the higher the indicator. The HHI used for solid wood products calculates the sum of squares of the raw material item's percentage of the total. If the HHI shows <0.10, it means that the supply items are diverse, and the market is more scattered; 0.10 < HHI <

0.18 means the item's diversity and market are moderately concentrated. If HHI is > 0.18, this market is over-concentrated and the item's supply lacks diversity [8,22].

The HHI model first squares the ratio of the import quantity of various solid wood products in each importing source country to the total amount of various solid wood products in all countries ($P_i{}^2$), and then calculates the total score by adding up the score of each of the 6 major items annually. Each year gives 7 HHI values, including the sum of 6 solid wood products and all raw materials. The formula is as below:

$$\mathrm{HHI} = \sum_{i=1}^{n} P_i{}^2 \qquad (2)$$

where $P_i$ = the proportion of solid wood products in total imports.

## 3. Results and Discussion

### 3.1. Quantity of Imports of Taiwan's Solid Wood Products

Based on the statistical data of import and export records from the customs administration under the Ministry of Finance, Figure 1 shows that the imports of solid wood products in the past ten years (2009–2018) were around $4500 \times 10^3$ to $6000 \times 10^3$ m$^3$, except for 2009, when there was less than $4000 \times 10^3$ m$^3$. The highest import quantity occurred in 2010, with $5636 \times 10^3$ m$^3$, and the average annual imports were $5264 \times 10^3$ m$^3$. Regarding different types of solid wood forest products, the import quantity of wood chips and pellets is the highest, accounting for 32% of the total import value of solid wood forest products. The average annual import quantity was $1672 \times 10^3$ m$^3$, followed by wood-based panels (with an average annual import quantity of $1475 \times 10^3$ m$^3$) and timber (with an average annual import quantity of $1157 \times 10^3$ m$^3$), which accounts for 28% and 22%, respectively, of the total import value of solid wood forest products.

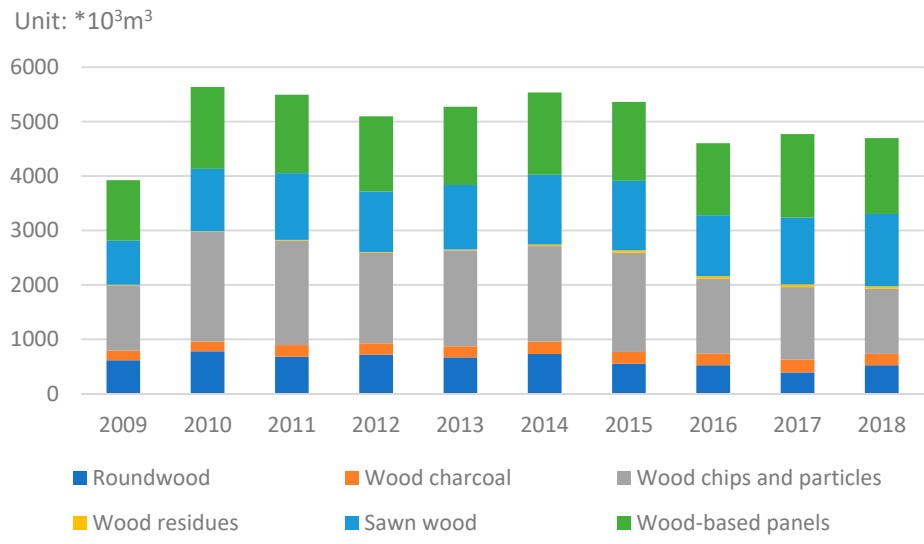

**Figure 1.** Quantity of Taiwan's major imported forest products, 2009–2018.

### 3.2. Taiwan's Major Importers of Solid Wood Products

In the past ten years (2009–2018), the overall import of solid wood products was sourced from 117 countries. In terms of categories, sawn wood is the highest, which was imported from 99 countries, followed by roundwood (93 countries), and wood chips and pellets the lowest (21 countries). According to the statistics, the total imports from the top 10 countries accounted for 88.12% of the total imports, which means that although the number of importing countries is quite diverse, supply is still concentrated in specific countries. Table 1 shows the top 10 countries, with the average total import quantity of solid wood products over the past ten years (2009–2018). The total import quantity of

each country, ranked from highest to lowest, begins with Australia, Malaysia, Thailand, Canada, and mainland China, down to Chile. In terms of types, roundwood is mainly imported from Malaysia, followed by New Zealand; wood charcoal is mainly imported from Vietnam, Indonesia and Malaysia; wood chips and pellets are mainly imported from Australia and Thailand, followed by Vietnam and Indonesia; wood residues (other wood chips, bark, sawdust) are mainly imported from Malaysia and Indonesia; sawn wood is mainly imported from Canada, Malaysia and the United States, followed by New Zealand, Australia and Vietnam; wood-based panels are mainly imported from Malaysia and mainland China, followed by Thailand and Indonesia. In terms of countries, Australia, which has the highest export quantity to Taiwan, mainly exports wood chips and pellets; Malaysia mainly exports roundwood and wood-based panels; Thailand mainly exports wood chips and pellets, and wood-based panels; Canada has the highest proportion of sawn wood exports, and China mainly exports wood-based panels.

**Table 1.** Average quantity of major imported forest products in Taiwan, by country.

| Countries | Roundwood | Wood Charcoal | Wood Chips and Pellets | Wood Residue | Sawn Wood | Wood-Based Panels | Total |
|---|---|---|---|---|---|---|---|
| Quantities of importing countries | 93 | 25 | 21 | 33 | 99 | 69 | 117 |
| Australia | 1.36 | – | 840.50 | 0.37 | 74.93 | 5.60 | 922.77 |
| Malaysia | 306.82 | 53.79 | 0.01 | 9.60 | 155.87 | 388.50 | 914.59 |
| Thailand | 0.02 | 14.84 | 411.24 | 0.63 | 3.20 | 223.56 | 653.48 |
| Canada | 16.44 | – | – | 0.24 | 388.35 | 16.82 | 421.86 |
| China | 0.42 | 1.99 | 1.07 | 3.75 | 17.86 | 351.51 | 376.61 |
| Indonesia | 0.77 | 66.21 | 155.62 | 7.70 | 4.38 | 107.35 | 342.03 |
| Viet Nam | 18.89 | 67.42 | 156.49 | 0.78 | 55.45 | 30.72 | 329.75 |
| New Zealand | 87.44 | – | – | 0.19 | 95.29 | 33.32 | 216.24 |
| United States of America | 16.50 | 0.22 | 0.06 | 1.00 | 153.96 | 4.64 | 176.36 |
| Chile | 0.07 | – | 30.40 | – | 42.60 | 14.58 | 87.64 |
| Total quantity of the first 10 countries | 448.73 | 204.47 | 1595.39 | 24.26 | 991.89 | 1176.60 | 4441.33 |
| Total quantity of all countries | 617.13 | 208.55 | 1605.45 | 26.92 | 1172.77 | 1409.15 | 5039.98 |
| Proportion of top 10 countries (%) | 72.71 | 98.04 | 99.37 | 90.12 | 84.58 | 83.50 | 88.12 |

Unit: $10^3$ m$^3$.

### 3.3. Result of Assessment Model

#### 3.3.1. SWI Assessment Results

SWI is used to measure the diversity and uniformity of products. A higher SWI value indicates higher diversity, which means improved wood supply safety, while a lower value represents low diversity of wood supply, so wood supply security is poor. Figure 2 shows the calculation of the SWI indicators for the data from 2009 to 2018. The results show that the SWI indicator of the three items of wood and wood-based panels is significantly higher than the other 2 types of solid wood products.

In this 10-year period (2009–2018), the SWI of the entire roundwood import quantity ranged from 1.31 to 2.06, and the trend gradually increased over time, with an average of 1.75 (Table 2). The trend increased visibly, starting from 2013, which indicates that the supply diversity of roundwood continues to rise. There are more importing countries, and the proportion of imports in each country is more balanced. The SWI of wood charcoal imports was between 1.25 and 1.50. Other than the increase in 2014, it showed a downward trend, with an average of 1.36. The supply diversity was roughly flat. The changes in wood chip and pellet imports were small, with the SWI maintaining at 0.75–1.33, but falling sharply to 0.8 and 0.75 in 2016 and 2017, indicating the declining supply diversity; in 2018,

it rebounded to 1.07, with an average of 1.10, which indicates that the supply diversity has declined overall in the past three years. The SWI of wood residues ranged from 1.45 to 2.32, with an average of 1.81, which had risen roughly from 2010 to 2013 and exceeded 2, while the rest of the years showed little change, but there has been another decline since 2014, and the supply diversity has changed drastically. The SWI for sawn wood imports was in the range of 2.09–2.52, maintaining an average of 2.0 and above. Except for a decline in 2012, the general trend was upward, with an average value of 2.29. It rose slightly above 2.4 starting from 2016. The SWI for wood-based panels ranged from 1.88 to 2.29. In 2009 and 2010, the value declined to below 2.0, and then rebounded, with an average of 2.11, showing fluctuations. The total ratio of the 6 products, when added up, was 2.29 to 2.82, with an average of 2.53, and the lowest point in 2010, but this was followed by a slight increase to reach the highest point in 2018. The overall supply diversity shows a slight upward trend. Based on this product importation data, it is clear that the higher the SWI, the higher the product supply diversity and security. The data indicate that as the SWI increases, the diversity of wood sources also increases with time.

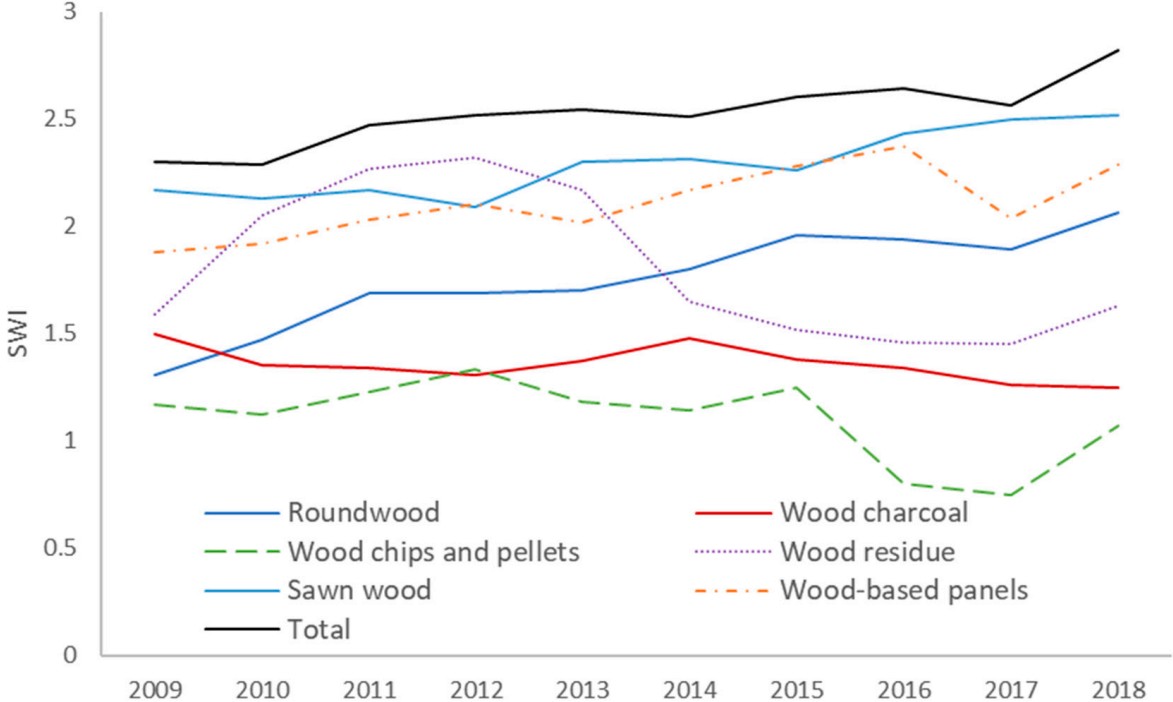

**Figure 2.** By quantity, SWI analysis of Taiwan's major imported forest products, 2009–2018.

**Table 2.** Major imported forest product SWI analysis.

| Year | Roundwood | Wood Charcoal | Wood Chips and Pellets | Wood Residue | Sawn Wood | Wood-Based Panels | Total |
|---|---|---|---|---|---|---|---|
| 2009 | 1.31 | 1.50 | 1.17 | 1.59 | 2.17 | 1.88 | 2.30 |
| 2010 | 1.47 | 1.35 | 1.12 | 2.05 | 2.13 | 1.92 | 2.29 |
| 2011 | 1.69 | 1.34 | 1.23 | 2.27 | 2.17 | 2.03 | 2.47 |
| 2012 | 1.69 | 1.31 | 1.33 | 2.32 | 2.09 | 2.10 | 2.52 |
| 2013 | 1.70 | 1.37 | 1.18 | 2.17 | 2.30 | 2.02 | 2.54 |
| 2014 | 1.80 | 1.48 | 1.14 | 1.65 | 2.31 | 2.17 | 2.51 |
| 2015 | 1.96 | 1.38 | 1.25 | 1.52 | 2.26 | 2.28 | 2.60 |
| 2016 | 1.94 | 1.34 | 0.80 | 1.46 | 2.43 | 2.37 | 2.64 |
| 2017 | 1.89 | 1.26 | 0.75 | 1.45 | 2.50 | 2.04 | 2.56 |
| 2018 | 2.06 | 1.25 | 1.07 | 1.63 | 2.52 | 2.29 | 2.82 |
| Average | 1.75 | 1.36 | 1.10 | 1.81 | 2.29 | 2.11 | 2.53 |

### 3.3.2. HHI Assessment Results

Figure 3 shows the calculation results of the HHI assessment. HHI reflects the concentration of importing countries and the total imports of wood products. For the 6 wood products in the HHI diversity assessment, the HHI value is the sum of the squares of the import ratios, ranging from 0 to 1. In special cases, the value 1 indicates absolute market concentration, and only one supplier contributes to the total import quantity. The lower the HHI value, the more diverse and secure the supply, which is roughly opposite to the SWI results. HHI is more risk-oriented; if HHI is < 0.10, it means that the supply items are diverse, and it is more scattered and sounder from the market perspective; if 0.10 < HHI < 0.18, the product diversity and market show concentration; if HHI is > 0.18, the market is too concentrated, and the supply item lacks diversity [8,22].

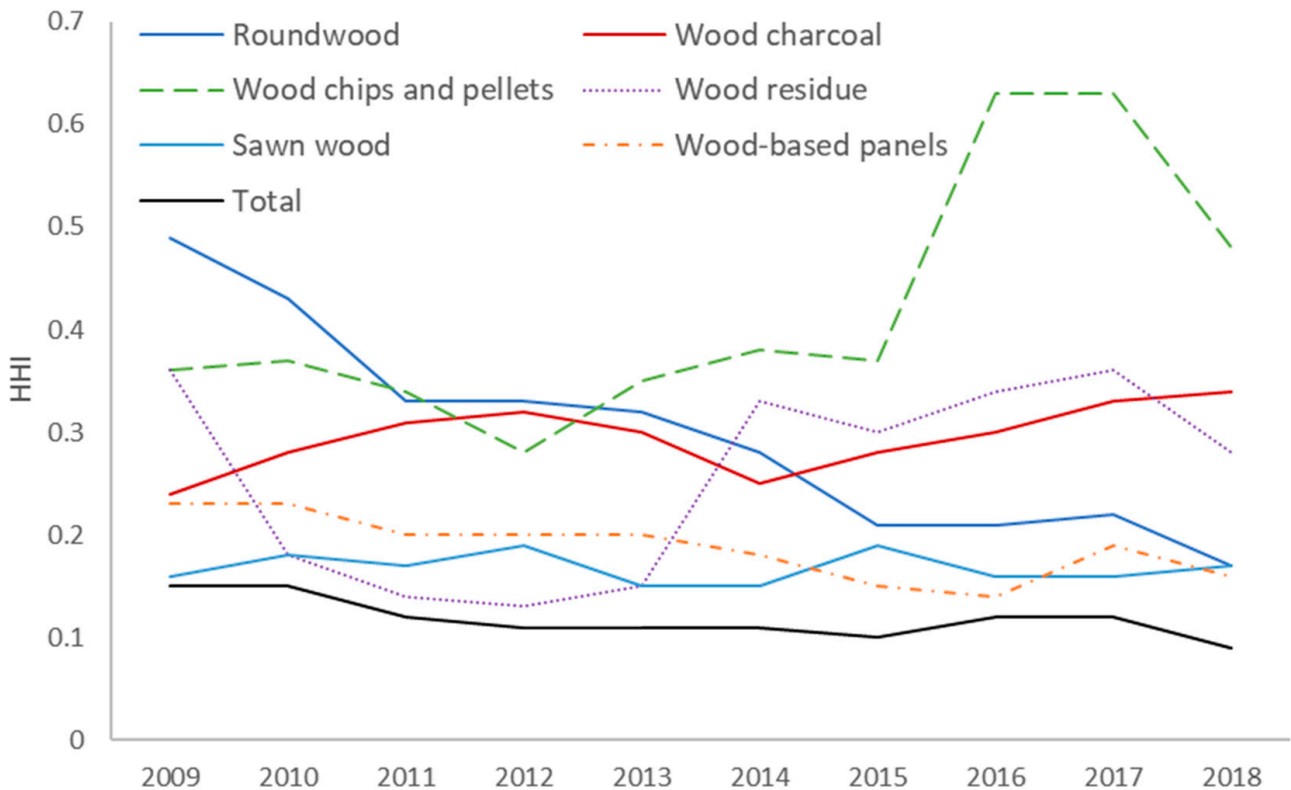

**Figure 3.** Quantity HHI analysis of Taiwan's major imported forest products, 2009–2018.

Based on the average, the HHI of the entire solid wood products is 0.12, which is between 0.10 < HHI < 0.18, meaning diversity and market concentration. Among the 6 products (other than sawn wood), HHI > 0.18 shows the excessive concentration of the market, especially among wood chips and pellets, with an average of HHI as high as 0.42. From the trend shown in Table 3 and Figure 3, the HHI of wood chips and pellets has increased, especially in 2016, and the supply tended to be exclusive. Roundwood shows the most obvious declining trend, from about 0.49 to 0.17, indicating that the roundwood market tends to be scattered and diverse; other items all show a slightly declining trend. Overall, sawn wood and wood-based panels are the most scattered and diverse items, and their HHI values remain significantly lower. For those with significant changes in HHI, the products such as roundwood, wood charcoal, wood chips and pellets, and wood residues are most noteworthy. The HHI value of wood chips and pellets, in particular, fluctuates greatly and requires more observation and adjustment.

**Table 3.** HHI analysis of major imported forest products.

| Year | Roundwood | Wood Charcoal | Wood Chips and Pellets | Wood Residue | Sawn Wood | Wood-Based Panels | Total |
|------|-----------|---------------|------------------------|--------------|-----------|-------------------|-------|
| 2009 | 0.49 | 0.24 | 0.36 | 0.36 | 0.16 | 0.23 | 0.15 |
| 2010 | 0.43 | 0.28 | 0.37 | 0.18 | 0.18 | 0.23 | 0.15 |
| 2011 | 0.33 | 0.31 | 0.34 | 0.14 | 0.17 | 0.20 | 0.12 |
| 2012 | 0.33 | 0.32 | 0.28 | 0.13 | 0.19 | 0.20 | 0.11 |
| 2013 | 0.32 | 0.30 | 0.35 | 0.15 | 0.15 | 0.20 | 0.11 |
| 2014 | 0.28 | 0.25 | 0.38 | 0.33 | 0.15 | 0.18 | 0.11 |
| 2015 | 0.21 | 0.28 | 0.37 | 0.30 | 0.19 | 0.15 | 0.10 |
| 2016 | 0.21 | 0.30 | 0.63 | 0.34 | 0.16 | 0.14 | 0.12 |
| 2017 | 0.22 | 0.33 | 0.63 | 0.36 | 0.16 | 0.19 | 0.12 |
| 2018 | 0.17 | 0.34 | 0.48 | 0.28 | 0.17 | 0.16 | 0.09 |
| Average | 0.30 | 0.30 | 0.42 | 0.26 | 0.17 | 0.19 | 0.12 |

*3.4. Summary*

According to the results, wood chips and pellets had the lowest average performance in SWI and offered the highest average HHI value but had the highest average annual import quantity, which indicates that wood chips and pellets are the riskiest items among the 6 solid wood products in Taiwan. On the other hand, while the sawn wood remains the highest average SWI value and the lowest average in HHI value, it contains the smallest supply risk. This assessment can be used as a reference for the development of national wood supply and demand strategies.

**4. Conclusions**

The solid wood product supplies of Taiwan are more diverse regarding wood chips and pellets, sawn wood, and wood-based panels. The first two products show an increasing trend in diversity, while there has been a decline in the supply diversity of wood residue and wood charcoal. However, the import diversity of roundwood has risen significantly in the past 10 years. Overall, the import diversity of solid wood products is gradually increasing; the number of imports representing the sources of importing countries has increased by approximately 36.78%, but the increase is only marginal. Basically, there has been little change in the past 10 years. HHI analysis presents slightly different results from that of SWI. The square sum of the import ratio of roundwood, wood charcoal, wood chips and pellets shows a declining trend, which indicates that in the early years, these import sources were mainly concentrated in several countries, and the import quantity from single sources is also large. Later, import quantity from a single country gradually decreased, but the number of countries of origin increased, so the trends in SWI slowly soared while HHI declined. Given these data, the difference in the meanings of the two assessment models is clear.

Although the countries of origin of Taiwan's solid wood product imports have increased slightly overall, the gaps of proportions between different imported items have been closer, and the diversity of wood supply has increased slightly, it is still difficult to say that the diversity of wood resources is optimistic based on the results of such data analysis. The fundamental problems still exist, such as the self-sufficiency rate of wood supply. If the self-sufficiency rate is increased, the resource autonomy can be further improved, and the supply risk can be more effectively reduced. Currently, the National Agriculture Congress calls for an increase in the wood self-sufficiency rate to 5% in the next 10 years. In addition to apportioning the proportion of importing countries, Taiwan must also move toward more autonomous goals.

The risk and diversity of solid wood products revealed from the assessment of these two models indicates that roundwood is currently the most stable product to increase the supply of diversity and reduce the risk. In addition, the performance of wood charcoal and wood chips and pellets is markedly better but has been weakening in the past 3 years.

The other products have changed little in the past 10 years, although the supply diversity of some has performed better than before, but others still see no progress, such as sawn wood and wood-based panels. These two models discuss only the diversity and security of imports. As far as the current situation is concerned, Taiwan still needs to import quantities of solid wood products in the future. To improve the structure of the supply chain, local wood production requires forestry policies to support the legality of logging in productive state-owned forests. The long-term shackles of the logging ban policy need to be removed as soon as possible to improve the domestic wood material supply structure, and to increase the wood self-sufficiency rate. Future related research can include a self-sufficiency rate calculation on the basis of these two models, to discuss the effect of raising self-sufficiency rate on the supply security of solid wood products, and whether the imports of wood products are only ensuring domestic demand or whether intended for export orientation.

**Author Contributions:** Data curation, M.-S.W.; writing—original draft preparation, J.-C.L.; writing—review and editing, W.-H.C.; project administration, C.-R.C. All authors have read and agreed to the published version of the manuscript.

**Funding:** This research received no external funding.

**Institutional Review Board Statement:** Not applicable.

**Informed Consent Statement:** Not applicable.

**Data Availability Statement:** Not applicable.

**Conflicts of Interest:** The authors declare no conflict of interest.

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
