# Peer review of "Security Assessment of Taiwan Solid Wood Product Supply"

_sustainability, doi:10.3390/su13095292_

Round 1

Reviewer 1 Report

This is a very interesting subject and the way it was delivered was remarkable. I always appreciate when methods used in other fields are used in the field of wood products. In this case, using methods from the energy field to studying wood products has been done objectively and the results were straightforward and gave a clear vision of the current context in Taiwan and the actions needed forward.

My observation are more about form than content:

Line 66 - Suggest replacing "we" with "Taiwan" to comply with objectivity.

Lines 105 - 119 - Delete template guidelines

Through the paper, it is clear that DIVERSITY is referred to as sources by country.   This is not clearly stated in the document, it needs a definition.  When describing wood products, there is a tendency to assume that diversity refers to species richness, for this reason, a clear statement in the introduction would be helpful to the readers

Figure 2 and Figure 3.  Use the same type/color of lines.

Author Response

  1. Line 66 - Suggest replacing "we" with "Taiwan" to comply with objectivity.

Revised as suggested

  1. Lines 105 - 119 - Delete template guidelines

Revised as suggested

  1. Through the paper, it is clear that DIVERSITY is referred to as sources by country.   This is not clearly stated in the document, it needs a definition.  When describing wood products, there is a tendency to assume that diversity refers to species richness, for this reason, a clear statement in the introduction would be helpful to the readers

Revised as suggested in the introduction. (lines 73-74)

  1. Figure 2 and Figure 3.  Use the same type/color of lines.

Revised as suggested

Reviewer 2 Report

The topic of the article is relevant at the national level and is worth examining, as the methodological approach can be used by other authors. The language of instruction is consistent, clear, and easy to read.

However, there are some shortcomings in the work that are suggested to be addressed before the publication of this article. It is recommended to strengthen the part of the literature analysis by using more scientific sources, presenting the advantages and main limitations of the methods used, the comparison with the works of other authors, if any. The level of investigation of the problem is not shown, so it is not clear whether other Taiwanese researchers have already conducted research in this area, what the authors' contribution is to the subject, and so on.

Some technical adjustments are also required. Section 2. Materials and Methods contains technical information-instructions for authors and has to be revised. The tables in the text splitted into two pages, it should be formatted to be on the same sheet. Hence the need for layout adjustments. The authors singled out the top15 countries from which timber and timber products are imported, but Table 1 shows that nine countries account for a significant share of total timber product imports, accounting for about 86.4% of total imports, so it is suggested to refine the description and focus on TOP9, or at least TOP10. Section 3 ends with a table and a figure, which also needs to be corrected. The relevant section should end with a summary of the results or the like. the scientific text recommends the use of a neutral writing style.

In the development of further research, it should be analyzed whether the imports of the goods under analysis are used, whether this only ensures domestic demand, or whether export orientation is observed, especially for higher value-added products. This is an important aspect that would allow a comprehensive assessment of the problem at hand. The study presented shows only one side of the coin. 

Author Response

  1. However, there are some shortcomings in the work that are suggested to be addressed before the publication of this article. It is recommended to strengthen the part of the literature analysis by using more scientific sources, presenting the advantages and main limitations of the methods used, the comparison with the works of other authors, if any. The level of investigation of the problem is not shown, so it is not clear whether other Taiwanese researchers have already conducted research in this area, what the authors' contribution is to the subject, and so on.

Response 1: Revised as suggested in the introduction. (lines 72-74 and 103-107)

  1. Some technical adjustments are also required.Section 2. Materials and Methods contains technical information-instructions for authors and has to be revised. The tables in the text splitted into two pages, it should be formatted to be on the same sheet. Hence the need for layout adjustments. The authors singled out the top15 countries from which timber and timber products are imported, but Table 1 shows that nine countries account for a significant share of total timber product imports, accounting for about 86.4% of total imports, so it is suggested to refine the description and focus on TOP9, or at least TOP10. Section 3 ends with a table and a figure, which also needs to be corrected. The relevant section should end with a summary of the results or the like. the scientific text recommends the use of a neutral writing style.

Response 2: We deleted the instructions for authors and modified the layout of all tables in this article. And refine the description of table 1 to focus on TOP10 countries.(lines 179-203). And we add the section of summary of results as section 3.4 (lines 276-283)

  1. In the development of further research, it should be analyzed whether the imports of the goods under analysis are used, whether this only ensures domestic demand, or whether export orientation is observed, especially for higher value-added products.This is an important aspect that would allow a comprehensive assessment of the problem at hand. The study presented shows only one side of the coin. 

Response 3: Thanks for the suggestion. We will consider the relevant questions in further study. And we mentioned this issue in the conclusion section (lines 326-328)

Round 2

Reviewer 2 Report

Thank you for the improvements